# Role of Allergen Immunotherapy in Asthma Treatment and Asthma Development

**Kazuyuki Nakagome** [1,2,*] and **Makoto Nagata** [1,2]

1   Department of Respiratory Medicine, Saitama Medical University, Saitama 350-0495, Japan; favre4mn@saitama-med.ac.jp
2   Allergy Center, Saitama Medical University, Saitama 350-0495, Japan
*   Correspondence: nakagomek-tky@umin.ac.jp; Tel.: +81-49-276-1319

**Abstract:** Allergen immunotherapy may modify the natural course of allergic diseases and induce remission. It includes subcutaneous immunotherapy (SCIT) and sublingual immunotherapy (SLIT). For asthma, allergen immunotherapy using house dust mite (HDM) improves clinical symptoms and airway hyperresponsiveness and decreases drug requirements. Furthermore, it has been suggested that allergen immunotherapy also has the following effects: (1) the effect can be maintained for more than a year even if the treatment is terminated, (2) the remission rate of childhood asthma can be increased, (3) new allergen sensitization can be suppressed, and (4) asthma development can be prevented if allergen immunotherapy was performed in the case of pollinosis. Allergen immunotherapy differs from conventional drug therapy, in particular the effect of modifying the natural course of allergic diseases and the effect of controlling complicated allergic diseases such as rhinoconjunctivitis. The general indication for HDM-SCIT in asthma is HDM-sensitized atopic asthma with mild-to-moderate disease and normal respiratory function. HDM allergens should be involved in the pathogenesis of asthma, and a duration of illness of less than 10 years is desirable. HDM-SLIT is available for allergic rhinitis but not for asthma in Japan. However, as the efficacy of SLIT for asthma has been fully proven internationally, SLIT is also applied in asthmatics with complicated allergic rhinitis in Japan.

**Keywords:** allergen immunotherapy; bronchial asthma; subcutaneous immunotherapy; sublingual immunotherapy

---

## 1. Introduction

Bronchial asthma has become a well-controlled disease in general because of advances in drug therapy centered on inhaled corticosteroid (ICS). However, ICS does not modify the natural course of asthma and is being positioned as a so-called symptomatic treatment [1,2]. Furthermore, ICS does not provide therapeutic benefits for allergic rhinoconjunctivitis, which is often complicated in asthmatic patients. Allergen immunotherapy is the only existing treatment that can be expected to induce immunological remission, that is, a possible cure of allergic diseases [3]. Moreover, allergen immunotherapy has therapeutic potency for a variety of allergic diseases simultaneously observed in an allergic individual. This treatment includes subcutaneous immunotherapy (SCIT) and sublingual immunotherapy (SLIT). Immunotherapy differs from drug therapy in that it may modify the natural course of allergic diseases by targeting allergen-specific Th2-type immune responses. In this article, we review the effect of allergen immunotherapy and its role in treating bronchial asthma. We also describe the current status of allergen immunotherapy for asthma in Japan.

## 2. SCIT in Bronchial Asthma

In asthma, meta-analyses have demonstrated that SCIT improves clinical symptoms and airway hyperresponsiveness and decreases drug requirements [4,5]. For example, Abramson et al. reported that the odds ratio for symptom improvement by SCIT with any allergen was 3.2 (95% CI 2.2–4.9), the odds ratio for drug reduction in SCIT using house dust mite (HDM) was 4.2 (95% CI 2.2–7.9), and the odds ratio for improvement of airway hypersensitivity was 6.8 (95% CI 3.8–12.0) [4].

The effect of the addition of SCIT with HDM (HDM-SCIT) to the guideline treatment was reported in patients with mild or moderate HDM-sensitized asthma [6]. In the immunotherapy group, a decrease in the frequency of inhalational β2-agonists and a significant improvement in peak flow were observed. Furthermore, in pediatric asthma, adding HDM-SCIT to the guideline treatment reduces the requirement for ICS and improves the morning peak flow [7]. Therefore, HDM-SCIT has an additional effect even after the standard treatment is already performed. Furthermore, as described below, allergen immunotherapy has a controlling effect on other allergic diseases such as rhinoconjunctivitis, which is often complicated in asthma, a maintenance effect for more than a year even after discontinuation of treatment, and may have an inhibitory effect on sensitization to new allergens. Therefore, allergen immunotherapy has shown significantly different clinical meaning from drug therapy represented by ICS.

However, the efficacy and effectiveness of SCIT in asthma remain controversial. Current evidence is derived from several small randomized controlled trials (RCTs), not only registration trials. Additionally, even in registration RCT, the effect of SCIT seems to be small to moderate. Furthermore, several biases including potential publication bias are also suggested in the meta-analyses [5].

Nonetheless, the United States adult asthma management guideline (EPR3) states that SCIT should be considered for allergic asthma in steps two to four (mild persistent-moderate persistent equivalent) of the six treatment steps [8]. The European Academy of Allergy and Clinical Immunology (EAACI) guideline states that HDM-SCIT is recommended as an add-on to regular asthma therapy for adults with controlled or partially controlled HDM-driven allergic asthma [9].

## 3. Modification of Natural Course of Allergic Diseases

Unlike other drug therapies, allergen immunotherapy may have the action of modifying the natural course of allergic diseases. Allergen immunotherapy remains effective for more than a year even after the end of treatment. For example, 3-year allergen immunotherapy for rhinitis/conjunctivitis improves symptoms and suppresses conjunctiva-induced allergic responses for 7 years after treatment discontinuation [10]. Furthermore, Durham et al. conducted RCTs and reported that 3-year allergen immunotherapy by SLIT results in a symptom-relieving effect for 1 or 2 years after treatment discontinuation [11,12].

The effect of inducing asthma remission in childhood asthma has also been reported. Allergen immunotherapy in pediatric patients with allergic rhinitis/asthma increases the rate of asthma remission after 5 years of treatment, and the remission can be maintained for 5 years after discontinuation [13].

Generally, allergen sensitization annually increases in patients with allergic asthma. However, allergen immunotherapy may have the long-term clinical effect of suppressing the spread of new allergen sensitizations. In a 15-year observational study, Marogna et al. reported that all allergic patients enrolled in the study and treated only by drug therapies were further sensitized with one or more new allergens in 15 years (100%). However, 3-, 4-, and 5-year allergen immunotherapy reduced the frequency of new allergen sanitization to 21%, 13%, and 12%, respectively [14].

Furthermore, allergen immunotherapy is effective in preventing asthma development in children with hay fever. In a 3-year observation study in children aged 6–14 years with rhinitis due to hay fever, 32 of 72 children in the control group developed asthma, whereas 19 of 79 children in the immunotherapy group developed asthma. There was a significantly lower rate of asthma development in the immunotherapy group (odds ratio 2.52; $p < 0.05$) [15]. Furthermore, this preventive effect

was maintained 7 years after the termination of immunotherapy. This study shows that allergen immunotherapy in patients with rhinitis may be effective in reducing the risk of developing asthma.

The modifying effect of allergen immunotherapy on the natural course of allergic diseases remains controversial. Long-term effects were not observed in all patients. The strongest evidence for long-term effects was obtained from a follow-up study after RCT by Durham et al., as described above. However, the efficacy of 3-year allergen immunotherapy using SLIT was assessed for only 2 years after discontinuation [12]. Two-year allergen immunotherapy using SLIT did not improve nasal response to allergen challenge at 1 year after discontinuation [16]. Concerning the prevention of new allergen sensitization, Di Bona et al. systemically reviewed the effect on developing new allergen sensitization and reported that the evidence was of a low-grade level and the risk of bias was high [17]. Small studies and studies with a shorter follow-up showed the highest benefit of allergen immunotherapy [17]. Moreover, a meta-analysis by Di Lorenzo et al. did not find evidence to support the preventing effect of new allergen sensitization in children [18]. Valovirta et al. reported that allergen immunotherapy using grass SLIT in patients with rhinoconjunctivitis without asthma suppressed the risk of experiencing asthma symptoms and using asthma medication, even up to 2 years after discontinuation. However, no effect was seen on the time to asthma onset [19]. Therefore, the actual effect of allergen immunotherapy on the modification of the natural course of allergic diseases must be further elucidated.

## 4. Allergen Immunotherapy in Asthma Patients with Rhinitis

In allergic rhinitis, allergen immunotherapy is already the standard treatment. Asthma has a very high rate of complication with rhinitis [20]. In patients with allergic rhinitis, nasal mucosal allergen exposure induces smooth muscle contraction and eosinophil infiltration in the lower respiratory tract and bronchial hypersensitivity [21]. In addition, in asthmatics without rhinitis, nasal mucosa has eosinophil infiltration, and direct administration of sensitized allergen into the trachea induces nasal inflammation and eosinophil infiltration into nasal mucosal tissues [22]. Therefore, in this manner, airway inflammation is worsened by nasal allergen exposure or nasal inflammation and vice versa, and the concept of "one airway, one disease" is well recognized [21].

In patients with asthma with allergic rhinitis, the treatment of rhinitis improves asthma symptoms and airway hyperresponsiveness and reduces asthma exacerbation [23]. According to the results of our questionnaire survey, patients with poor control of asthma symptoms are aware that asthma symptoms worsen as their nasal symptoms worsen, and asthma symptoms tend to improve with nasal treatment [24]. Therefore, the management of rhinitis is important for the treatment of asthma with rhinitis, and allergen immunotherapy is a reasonable strategy to control rhinitis and asthma. In clinical practice, comprehensive treatment should be considered for not only bronchial asthma but also complicated allergic diseases in individual asthmatics.

## 5. Selection of Asthma Patients for HDM-SCIT

The indication for HDM-SCIT in atopic asthma is mild-to-moderate persistent type with percent predicted forced expiratory volume in one second (%FEV$_1$) of ≥70%. Treatment should be started in the stable period. Generally, a strong effect can be expected in a patient who is not sensitized to other allergens and is sensitized to HDM alone. We observed that the clinical effect, based on the rate of obtaining step-down of asthma severity, is significantly lower in patients with >10 years of disease or forced expiratory volume in one second (FEV$_1$) of <70% [25]. Therefore, this therapy is more effective when applied as an intervention in the early phase of atopic asthma, in which airway remodeling has not developed. Moreover, as described above, in asthmatic patients with allergic rhinitis, an effect on rhinitis can be simultaneously expected.

Patients with heart, liver, kidney, thyroid, or collagen disease should be excluded, and therapy initiation during pregnancy should be avoided. However, treatment can be continued if the patient has already reached maintenance therapy before pregnancy. Furthermore, as the therapeutic effect of

ICS on asthma is diminished in smoking patients, it is assumed that the effect of immunotherapy in smoking patients is not sufficiently exerted.

## 6. Mechanisms of Allergen Immunotherapy

Allergen immunotherapy increases the concentrations of serum allergen-specific IgG or IgG4 and IgA antibodies (Abs) (Figure 1) [26–29] and transiently increases concentrations of allergen-specific IgE Abs. Several studies have demonstrated the inhibitory capacity of IgG or IgG4 for IgE-dependent immune responses. IgG or IgG4 can compete with IgE for allergen, inhibiting allergen-IgE complex formation [30]. Thus, it prevents cross-linking of high-affinity IgE receptors (FcεRI) on basophils and mast cells, which suppress histamine release, and blocks binding of allergen-IgE complexes to low-affinity receptors (FcγRIIb) on B cells [31], which suppress IgE-facilitated antigen presentation to T cells.

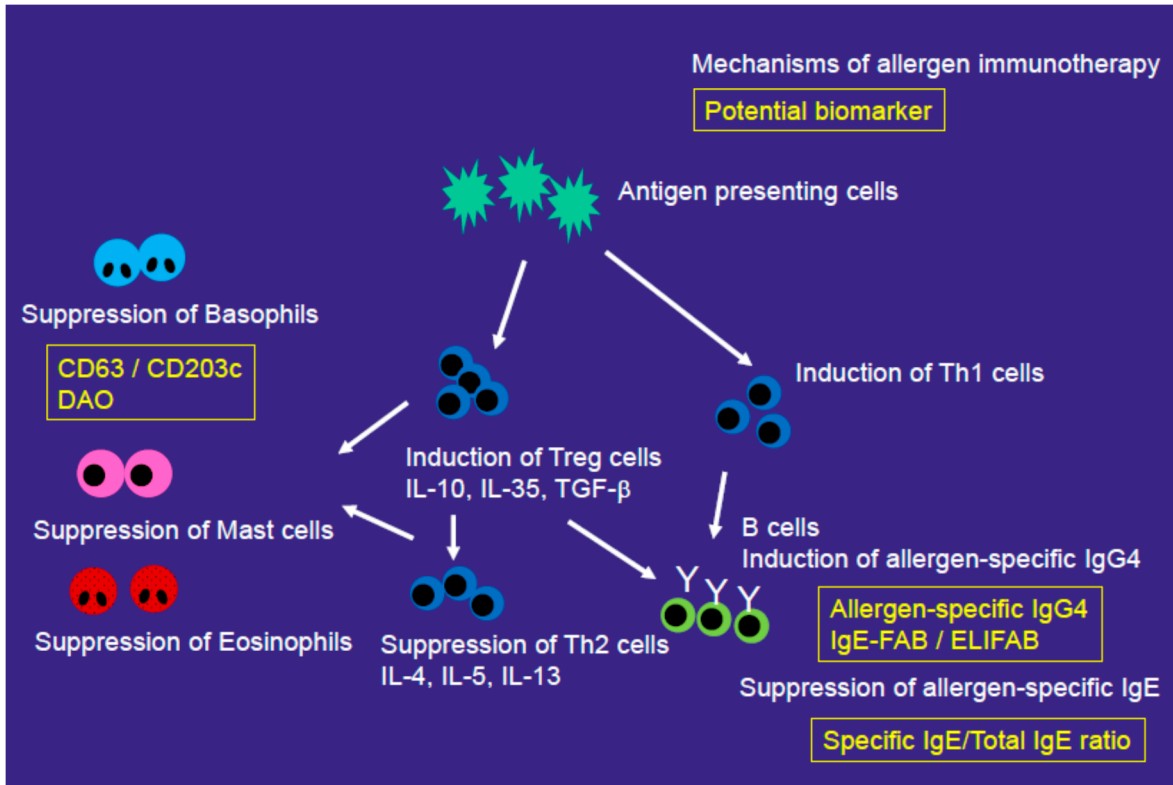

**Figure 1.** Mechanisms of allergen immunotherapy. Allergen immunotherapy induces T regs or Th1 cells and suppresses Th2 cells, eosinophils, mast cells, and basophils. It also induces allergen-specific IgG4 and suppresses allergen-specific IgE. Yellow text indicates potential biomarkers for allergen immunotherapy. DAO, diamine oxidase; ELIFAB, enzyme-linked immunosorbent facilitated antigen-binding; FAB, facilitated allergen binding.

Local production of Th2 cytokines, such as IL-4 and IL-5, or the numbers of Th2 cells are decreased by allergen immunotherapy (Figure 1) [32–34]. We found that immunotherapy attenuates HDM-specific production of thymus and activation-regulated chemokine, a potent chemokine activator of Th2 cells, from peripheral blood mononuclear cells (PBMCs) obtained from patients with HDM-sensitized allergic asthma, suggesting that immunotherapy can reduce accumulation of Th2 cells during allergen exposure [3]. Furthermore, immunotherapy suppresses allergen-induced Th2 cytokines such as IL-5 from PBMCs of allergic patients [35]. Therefore, immunotherapy can induce systemic immunological changes in response to allergens and provides some clinical benefits in allergic asthma. In addition to the effects on Th2-mediated immune responses, allergen immunotherapy induces regulatory T cells (Tregs) (Figure 1) [27,29,36–38]. Tregs are divided into two subsets: natural regulatory T cells (nTregs),

which express the transcription factor forkhead box P3 (FOXP3), and inducible regulatory T cells (iTregs), which produce IL-10, IL-35, and TGF-β. For example, allergen immunotherapy increases local FOXP3[+] T cells [36,37]. Allergen immunotherapy also increases local IL-10-expressing T cells [27,38] and TGF-β+ T cells [29]. However, the involvement of Tregs in allergen immunotherapy for Th2 suppression is probably regulated by multiple factors, including allergen and time of assessment. Furthermore, the role of regulatory B cells (Bregs), which also produce IL-10 and can suppress immune responses, has also been highlighted [39,40]. In bee venom-tolerant patients, IL-10-producing B cells, which express CD25 and CD71 but not CD73, are increased and associated with IgG4 production [39,40]. In addition to IL-10, Bregs reveal their suppressive property through TGF-β and IL-35 [39]. Moreover, allergen immunotherapy increases allergen challenge-induced expression of IL-12 mRNA in the skin [41]. These findings suggest that allergen immunotherapy suppresses T cell-mediated airway inflammation via downmodulation of Th2 cells and induction of Tregs or Th1 cells. As for type-2 innate lymphoid cells (ILC2), which are also important sources of Th2 cytokines such as IL-5 and IL-13, allergen immunotherapy decreases the number of ILC2 in peripheral blood [42], although this finding is controversial [43].

Eosinophilic airway inflammation is an important feature of bronchial asthma. Infiltration of activated eosinophils in the airways is associated with asthma severity. Allergen immunotherapy suppresses airway inflammation, including the numbers of infiltrated eosinophils and concentrations of eosinophil-specific granule proteins (Figure 1) [3]. For circulating eosinophils to accumulate in asthmatic airways, they must adhere to and then migrate across vascular endothelial cells. These processes are largely regulated by cytokines/chemokines produced by various cells, including Th2 cells [44–46]. Increased adhesion of peripheral blood eosinophils and increased chemotactic activity of eosinophils into the airways are observed during the allergen exposure period in birch pollen asthma, and allergen immunotherapy suppresses increased eosinophil adhesion and chemotactic activity [3,32]. We reported that stimulation of PBMCs from HDM-sensitized allergic asthmatics with HDM increases eosinophil adhesion-inducing activity, eosinophil chemotactic activity, and eosinophil transendothelial migration-inducing activity, and the increase in these eosinophil activities was attenuated by allergen immunotherapy [3,47]. These findings suggest that the modification of Th2-mediated immune responses to specific allergens by allergen immunotherapy can suppress eosinophil accumulation in the airways.

## 7. Biomarker for Allergen Immunotherapy

Biomarkers to predict effects of allergen immunotherapy are clinically important. The EAACI Task Force reported a consensus statement on biomarkers of allergen immunotherapy [48,49]. Biomarkers were grouped into seven domains: (1) IgE (total IgE, specific IgE, and specific IgE/total IgE ratio), (2) IgG subclasses (allergen-specific IgG1 and specific IgG4, including the specific IgE/IgG4 ratio), (3) serum inhibitory activity for IgE (assessed by IgE-facilitated antigen-binding (IgE-FAB) assay), (4) basophil activation, (5) cytokines and chemokines, (6) cellular markers (Tregs, Bregs, and dendritic cells), and (7) in vivo biomarkers, which include provocation tests. Although the optimal biomarker for the prediction of the effects has not been identified, specific IgE/total IgE ratio, allergen-specific IgG4 including the specific IgE/IgG4 ratio, IgE-FAB, and basophil activation are thought to be potentially useful [48,49].

An early increase in specific IgE levels is observed during allergen immunotherapy, and the seasonal increase in IgE subsequently slows thereafter [28,50]. Then, specific IgE gradually decreases over several years [51], although there is no clear association between changes in specific IgE levels and the clinical response [29,38]. In contrast, several studies suggested that the ratio of specific IgE/total serum IgE at baseline correlates with clinical response to immunotherapy [52,53], although these findings were not reproduced in other studies [54,55].

Allergen-specific IgG subtypes including IgG4 are increased during allergen immunotherapy as compared with baseline values (Figure 1) [56–58]. We reported that there was a high correlation between the increase in log provocative doses causing a 20% decline in $FEV_1$ and increase in the ratio of

HDM-specific IgG4 to IgG1, suggesting that the increase in IgG4 is associated with the improvement of airway hyperresponsiveness in asthma [3]. However, a correlation between allergen IgG4 concentrations and clinical outcomes has not been reported in all studies [56,57,59]. For example, in a long-term follow-up study that evaluated patients up to 6 years after discontinuation of allergen immunotherapy, there was no correlation between allergen IgG4 concentrations and clinical outcomes [59]. As for specific IgE/IgG4 ratio, a decreased ratio was reported after allergen immunotherapy and associated with a reduction in late cutaneous skin reactions [48].

IgG-associated IgE-inhibitory activity can be assessed by IgE-FAB assay (Figure 1) [49,60]. The IgE-FAB assay measures the ability of IgG-containing serum obtained after allergen immunotherapy to inhibit FcεRII-dependent binding of allergen-IgE complexes to B cells, although this assay is currently confined to specialized laboratories [28,49]. Another approach is the enzyme-linked immunosorbent-facilitated antigen-binding (ELIFAB) assay [49,61]. Several studies suggest a good correlation between IgE-FAB and ELIFAB results and the clinical response to immunotherapy as compared with serum IgG or IgG4 levels [28,49,61]. This is probably because IgE-FAB or ELIFAB measures the function of affinity and/or avidity of Ab binding.

Basophil activation can be assessed by measuring the expression of surface markers, such as CD63 and CD203c, using whole blood. Although CD63 expression indicates basophil degranulation (Figure 1) [62], CD203c is a specific basophil marker that also indicates IL-3-dependent activation. Intracellular staining of phycoerythrin-conjugated diamine oxidase (DAO) has recently been highlighted (Figure 1) [63,64]. DAO binds to its substrate histamine, such that allergen stimulation reduces intracellular DAO levels in basophils proportional to the amount of intracellular histamine released. This reduction has been detected during both SCIT and SLIT [63,64].

## 8. Clinical Application of SLIT for Asthma

As SCIT has a risk of systemic reactions, although infrequent, SLIT has been developed mainly in Europe as a safer alternative method to SCIT.

It was reported from the 1990s that SLIT with HDM (HDM-SLIT) improves symptom scores and airway hyperresponsiveness in patients with HDM-sensitized allergic asthma [65]. Since then, in asthmatic patients with rhinitis due to hay fever, SLIT has been shown to improve asthma symptoms, decrease the use of bronchodilators, and improve respiratory function compared with symptomatic treatment. Furthermore, Marogna et al. compared the effects of SLIT and ICS in patients with mild asthma and concomitant rhinitis due to grass pollen [66]. After a run-in season, patients were randomized to either 800 μg/day budesonide, an ICS, during the pollen season or continuous grass SLIT for 5 years. Asthma symptoms significantly decreased in both groups; however, improvements were greater in the SLIT group at 3 and 5 years. Furthermore, a decrease in both nasal symptoms and nasal eosinophils was observed only in the SLIT group.

SLIT may be inferior to conventional SCIT in terms of efficacy [67–69]. For example, Di Bona et al. reported that SCIT is more effective than SLIT as assessed by symptom scores or medication scores [68]. However, because SLIT is less painful, convenient, and highly safe, it can be used in many countries in general clinical practice.

Recently, the HDM-SLIT tablet developed by the Danish ALK was reported to be effective in bronchial asthma in a large-scale clinical study [70,71]. This HDM-SLIT tablet (6 standardized quality (SQ)) significantly reduces ICS use in asthma (SLIT 42%, placebo 15%, $p = 0.0011$) [70]. Furthermore, this HDM-SLIT tablet [6SQ] suppresses moderate-to-severe asthma exacerbation associated with ICS reduction (hazard ratio 0.72, $p = 0.045$) [71]. Based on this evidence, the Global Initiative for Asthma includes the description that HDM-SLIT should be considered in adult HDM-sensitized patients with allergic rhinitis, provided that %$FEV_1$ is >70% [72].

### 9. Allergen Immunotherapy in Japan

The standardized HDM allergen for SCIT was not available in Japan until 2015. Before 2015, SCIT using house dust (HD), collected from the general house, was utilized as an alternative therapeutic agent. The main component of HD is mites, but there were problems with product quality, and it was necessary to improve the effect and safety by standardizing allergens. The standardized purified HDM allergen for SCIT was prepared in 2015 and is currently used for the treatment of asthma.

For SLIT, two HDM-SLIT tablets were approved for allergic rhinitis, but not for asthma, in 2015. The tablet developed by Torii Pharmaceutical Co., Ltd., Tokyo, Japan (MITICURE®) is the same as the tablet manufactured by ALK, in which the effect on asthma has been fully proven in Europe as described (10,000 Japanese allergy unit (JAU); the maintenance dose in Japan is equivalent to 6SQ in Europe).

Recently, the effect of the HDM-SLIT tablet on asthma has also been determined in Japan. In HDM-sensitized atopic asthma with rhinitis, the addition of MITICURE® to the standard treatment improved the symptom scores of asthma, fractional exhaled nitric oxide, $FEV_1$, and airway wall thickening in chest CT [73], suggesting that HDM-SLIT can suppress not only airway inflammation but also airway remodeling of asthma. Furthermore, a study of the effect of MITICURE® on asthma exacerbation associated with ICS reduction demonstrated that this treatment suppresses asthma exacerbation in patients who used short-acting β2-agonists during the observation period [74], which is consistent with the previous study in Europe [71].

Japanese cedar pollen (JCP) is widely scattered in the spring in Japan. Pollinosis by JCP is a representative seasonal rhinitis in Japan. People living in urban areas also suffer from pollinosis because JCP is scattered over tens of thousands of kilometers. One epidemiological study reported that the prevalence of Japanese cedar pollinosis was 26.5% [75]. Furthermore, its prevalence increased by about 10% in 10 years. This situation in Japan is unique in that Japanese cedars are planted forests and not natural ones. Similar to other pollens, JCP has been reported to exacerbate asthma. For example, Hojo et al. reported that the asthma control level, measured by the visual analog scale of Self-Assessment of Allergic Rhinitis and Asthma Questionnaire and Asthma Control Test score, worsened during the JCP-scattering season in asthmatic patients with allergic rhinitis by JCP, although 84% received treatment for rhinitis [76]. Asthma control during the pollen season was impaired in 18–38% of asthmatics with seasonal rhinitis with JCP [76]. Although the mechanisms for asthma exacerbation by JCP have not been fully clarified, several possible mechanisms are proposed. For example, fine orbicules (about 1 μm) adhering to the surface of JCP can reach the lower respiratory tract and directly induce asthma exacerbation. In addition, the effects of nasal obstruction, mediator released locally in the nose, and increased systemic cytokine production may be involved in JCP-related asthma exacerbation.

Regarding JCP-related asthma, we have confirmed that treatment with JCP-SLIT almost completely abrogates the appearance of asthma exacerbation during the JCP-scattering season [77], supporting the certain prevention effect of SLIT on asthma exacerbation. Collectively, these findings indicate that HDM- or JCP-SLIT should be considered for asthmatic patients with rhinitis.

### 10. Adherence to Allergen Immunotherapy

One important problem which should be addressed is adherence to allergen immunotherapy [78,79]. As described above, most studies reported that more than 3 years of treatment is required to exert a modifying effect on the natural course of allergic diseases. For example, Kiel et al. reported that only 23% and 7% of patients receiving SCIT and SLIT, respectively, continued treatment for 3 years [78]. Sena et al. reported poor adherence using manufacturer sales data: SLIT prescription sales decreased from 100% to 44%, 28%, and 13% in the first, second, and third years, respectively, suggesting that <20% of patients had good adherence after 3 years [79]. Therefore, barriers to allergen immunotherapy adherence and strategies to improve compliance must be further investigated.

In Japan, the treatment continuation rates of JCP-SLIT may be high. Yuta et al. reported that good adherence by direct calculation from prescription for 2 years was observed in 83% of patients [80]. We confirmed the similar treatment continuation rates of JCP-SLIT [81]. Although the reason of the difference is difficult to ascertain, the symptoms of patients with rhinitis by JCP were not sufficiently alleviated by drug treatment alone: consequently patients may become enthusiastic to receive JCP-SLIT. In our case, careful explanation on the role of SLIT and the importance of continuation for at least 3 years are usually performed. We assessed the predictors of adherence to JCP-SLIT prospectively and found that age younger than 40.5 years was the cutoff value for predicting poor adherence to JCP-SLIT [81]. Therefore, to reduce the discontinuation rate, the necessity of long-term treatment continuity should be clearly communicated prior to commencing treatment, especially for patients younger than 40 years.

## 11. Conclusions

In HDM-sensitized asthma, HDM-SCIT improves clinical symptoms and airway hyperresponsiveness and decreases drug requirements. Furthermore, HDM- or JCP-SLIT can decrease asthma exacerbation and drug requirements. Current pharmacotherapy, such as ICS, provides powerful anti-symptomatic benefits in asthma; however, it does not modify the natural course of allergic diseases. In contrast, allergen immunotherapy targets the immunological background including the pathological activation of Th2 cells. Thus, it is expected to lead to long-term amelioration of asthma and allergic diseases. It is hoped that allergen immunotherapy is more widely applied in the treatment of asthma as a strategy for comprehensive management of allergy symptoms and modification of disease course.

**Author Contributions:** K.N. wrote the manuscript. M.N. edited the manuscript. All authors have read and agreed to the published version of the manuscript.

**Funding:** This work was supported by a grant from the Ministry of Education, Culture, Sports, Science and Technology (15K09228).

**Conflicts of Interest:** M.N. received honoraria from Torii Pharmaceutical Co., Ltd. K.N. has no conflict of interest.

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
