# Peer review of "Role of Allergen Immunotherapy in Asthma Treatment and Asthma Development"

_allergies, doi:10.3390/allergies1010003_

Round 1

Reviewer 1 Report

The reviews is well written and comprehensive. The main defect is that the the general tone of the review is too positive on AIT effects on clinical ground.

In reality, both the efficacy and the effectiveness of SCIT and SLIT are more controversial. The evidence come from several small RCTs, non only registration trials, and even in registration RCT the effect seems to be small to moderate. Moreover, observational studies are limited by several sources of bias that investigators did not control for.  

Furthermore, other effects such as prevention of new sensitizations or prevention of asthma are doubtful (see Di Bona et al, Allergy 2017; Valovirta et al. JACI 2018, respectively).

These studies were not cited by the Authors, but they are very useful to give the reader the opportunity to assess the importance of all the evidence, not only those favorable to AIT.

Same considerations are valid also for long-term effect, which was showed in some patients, but not in all. At this regard, the strongest evidence comes from the Durham follow-up study, which assessed the patients at one and two years from AIT interruption after a RCT. One year follow-up seems inadequate to answer this question.

Not to talk of the effectiveness. It was clearly shown by many reports that if AIT is not performed for at least 3 consecutive years, the short- and long-term  effects are abolished. Well, we know that SLIT is administered for 3 years in less than 10% of patients, whereas SCIT in 20% to 40% of patients.

This evidence dramatically change the general picture. The Authors did not confront this problem.

I suggest to change the tone of the review into a more dubitative one.

Add the following citations:

  1. Di Bona D, Plaia A, Leto-Barone MS, La Piana S, Macchia L, Di Lorenzo G. Efficacy of allergen immunotherapy in reducing the likelihood of developing new allergen sensitizations: a systematic review. Allergy. 2017 May;72(5):691-704. doi: 10.1111/all.13104. Epub 2017 Jan 23. PMID: 27926981.
  2. Di Lorenzo G, Leto-Barone MS, La Piana S, Plaia A, Di Bona D. The effect of allergen immunotherapy in the onset of new sensitizations: a meta-analysis. Int Forum Allergy Rhinol. 2017 Jul;7(7):660-669. doi: 10.1002/alr.21946. Epub 2017 May 23. PMID: 28544523.  
  3. Di Bona D, Plaia A, Leto-Barone MS, La Piana S, Di Lorenzo G. Efficacy of subcutaneous and sublingual immunotherapy with grass allergens for seasonal allergic rhinitis: a meta-analysis-based comparison. J Allergy Clin Immunol. 2012 Nov;130(5):1097-1107.e2. doi: 10.1016/j.jaci.2012.08.012. Epub 2012 Sep 27. PMID: 23021885.
  4. Dretzke J, Meadows A, Novielli N, Huissoon A, Fry-Smith A, Meads C. Subcutaneous and sublingual immunotherapy for seasonal allergic rhinitis: a systematic review and indirect comparison. J Allergy Clin Immunol. 2013 May;131(5):1361-6. doi: 10.1016/j.jaci.2013.02.013. Epub 2013 Apr 1. PMID: 23557834.
  5. Valovirta E, Petersen TH, Piotrowska T, Laursen MK, Andersen JS, Sørensen HF, Klink R; GAP investigators. Results from the 5-year SQ grass sublingual immunotherapy tablet asthma prevention (GAP) trial in children with grass pollen allergy. J Allergy Clin Immunol. 2018 Feb;141(2):529-538.e13. doi: 10.1016/j.jaci.2017.06.014. Epub 2017 Jul 6. PMID: 28689794.
  6. Scadding GW, Calderon MA, Shamji MH, Eifan AO, Penagos M, Dumitru F, Sever ML, Bahnson HT, Lawson K, Harris KM, Plough AG, Panza JL, Qin T, Lim N, Tchao NK, Togias A, Durham SR; Immune Tolerance Network GRASS Study Team. Effect of 2 Years of Treatment With Sublingual Grass Pollen Immunotherapy on Nasal Response to Allergen Challenge at 3 Years Among Patients With Moderate to Severe Seasonal Allergic Rhinitis: The GRASS Randomized Clinical Trial. JAMA. 2017 Feb 14;317(6):615-625. doi: 10.1001/jama.2016.21040. PMID: 28196255; PMCID: PMC5479315.
  7. Durham SR, Emminger W, Kapp A, de Monchy JG, Rak S, Scadding GK, Wurtzen PA, Andersen JS, Tholstrup B, Riis B, Dahl R. SQ-standardized sublingual grass immunotherapy: confirmation of disease modification 2 years after 3 years of treatment in a randomized trial. J Allergy Clin Immunol. 2012 Mar;129(3):717-725.e5. doi: 10.1016/j.jaci.2011.12.973. Epub 2012 Jan 29. PMID: 22285278.
  8. Senna G, Lombardi C, Canonica GW, Passalacqua G. How adherent to sublingual immunotherapy prescriptions are patients? The manufacturers' viewpoint. J Allergy Clin Immunol. 2010;126(3):668-669.
  9. Kiel MA, Röder E, Gerth van Wijk R, Al MJ, Hop WCJ, Rutten-van Mölken MPMH. Real-life compliance and persistence among users of subcutaneous and sublingual allergen immunotherapy. J Allergy Clin Immunol. 2013;132(2):353-60.

Author Response

Response to Reviewers’ comments

Reviewer #1

The reviews is well written and comprehensive. The main defect is that the the general tone of the review is too positive on AIT effects on clinical ground.

In reality, both the efficacy and the effectiveness of SCIT and SLIT are more controversial. The evidence come from several small RCTs, non only registration trials, and even in registration RCT the effect seems to be small to moderate. Moreover, observational studies are limited by several sources of bias that investigators did not control for. 

Furthermore, other effects such as prevention of new sensitizations or prevention of asthma are doubtful (see Di Bona et al, Allergy 2017; Valovirta et al. JACI 2018, respectively).

These studies were not cited by the Authors, but they are very useful to give the reader the opportunity to assess the importance of all the evidence, not only those favorable to AIT.

Same considerations are valid also for long-term effect, which was showed in some patients, but not in all. At this regard, the strongest evidence comes from the Durham follow-up study, which assessed the patients at one and two years from AIT interruption after a RCT. One year follow-up seems inadequate to answer this question.

Not to talk of the effectiveness. It was clearly shown by many reports that if AIT is not performed for at least 3 consecutive years, the short- and long-term effects are abolished. Well, we know that SLIT is administered for 3 years in less than 10% of patients, whereas SCIT in 20% to 40% of patients.

This evidence dramatically change the general picture. The Authors did not confront this problem.

I suggest to change the tone of the review into a more dubitative one.

Add the following citations:

Di Bona D, Plaia A, Leto-Barone MS, La Piana S, Macchia L, Di Lorenzo G. Efficacy of allergen immunotherapy in reducing the likelihood of developing new allergen sensitizations: a systematic review. Allergy. 2017 May;72(5):691-704. doi: 10.1111/all.13104. Epub 2017 Jan 23. PMID: 27926981.

Di Lorenzo G, Leto-Barone MS, La Piana S, Plaia A, Di Bona D. The effect of allergen immunotherapy in the onset of new sensitizations: a meta-analysis. Int Forum Allergy Rhinol. 2017 Jul;7(7):660-669. doi: 10.1002/alr.21946. Epub 2017 May 23. PMID: 28544523. 

Di Bona D, Plaia A, Leto-Barone MS, La Piana S, Di Lorenzo G. Efficacy of subcutaneous and sublingual immunotherapy with grass allergens for seasonal allergic rhinitis: a meta-analysis-based comparison. J Allergy Clin Immunol. 2012 Nov;130(5):1097-1107.e2. doi: 10.1016/j.jaci.2012.08.012. Epub 2012 Sep 27. PMID: 23021885.

Dretzke J, Meadows A, Novielli N, Huissoon A, Fry-Smith A, Meads C. Subcutaneous and sublingual immunotherapy for seasonal allergic rhinitis: a systematic review and indirect comparison. J Allergy Clin Immunol. 2013 May;131(5):1361-6. doi: 10.1016/j.jaci.2013.02.013. Epub 2013 Apr 1. PMID: 23557834.

Valovirta E, Petersen TH, Piotrowska T, Laursen MK, Andersen JS, Sørensen HF, Klink R; GAP investigators. Results from the 5-year SQ grass sublingual immunotherapy tablet asthma prevention (GAP) trial in children with grass pollen allergy. J Allergy Clin Immunol. 2018 Feb;141(2):529-538.e13. doi: 10.1016/j.jaci.2017.06.014. Epub 2017 Jul 6. PMID: 28689794.

Scadding GW, Calderon MA, Shamji MH, Eifan AO, Penagos M, Dumitru F, Sever ML, Bahnson HT, Lawson K, Harris KM, Plough AG, Panza JL, Qin T, Lim N, Tchao NK, Togias A, Durham SR; Immune Tolerance Network GRASS Study Team. Effect of 2 Years of Treatment With Sublingual Grass Pollen Immunotherapy on Nasal Response to Allergen Challenge at 3 Years Among Patients With Moderate to Severe Seasonal Allergic Rhinitis: The GRASS Randomized Clinical Trial. JAMA. 2017 Feb 14;317(6):615-625. doi: 10.1001/jama.2016.21040. PMID: 28196255; PMCID: PMC5479315.

Durham SR, Emminger W, Kapp A, de Monchy JG, Rak S, Scadding GK, Wurtzen PA, Andersen JS, Tholstrup B, Riis B, Dahl R. SQ-standardized sublingual grass immunotherapy: confirmation of disease modification 2 years after 3 years of treatment in a randomized trial. J Allergy Clin Immunol. 2012 Mar;129(3):717-725.e5. doi: 10.1016/j.jaci.2011.12.973. Epub 2012 Jan 29. PMID: 22285278.

Senna G, Lombardi C, Canonica GW, Passalacqua G. How adherent to sublingual immunotherapy prescriptions are patients? The manufacturers' viewpoint. J Allergy Clin Immunol. 2010;126(3):668-669.

Kiel MA, Röder E, Gerth van Wijk R, Al MJ, Hop WCJ, Rutten-van Mölken MPMH. Real-life compliance and persistence among users of subcutaneous and sublingual allergen immunotherapy. J Allergy Clin Immunol. 2013;132(2):353-60.

We thank the reviewer for these useful suggestions. Accordingly, we have changed the tone of the review into a more dubitative one, especially in the description of the effect of allergen immunotherapy on the modification of the natural course of allergic diseases. We added a critical review about the effect of allergen immunotherapy on the treatment of asthma or other allergic diseases and the effect of allergen immunotherapy on the development of new allergen sensitization in the revised manuscript. We also described the efficacy of SLIT compared with SCIT in the revised manuscript. Furthermore, we have added a new paragraph about allergen immunotherapy adherence, which is an important problem that affects its effectiveness, in the revised manuscript. We cited all papers the reviewer suggested in the revised manuscript.

Reviewer 2 Report

The manuscript “Allergen immunotherapy in asthma” addresses the effect of allergen immunotherapy and its role in treating bronchial asthma compared to with symptomatic treatments. The manuscript is in general well written, although some sentences need to be clarified/rephrased. The proposed manuscript would be of interest to Allergies journal after the authors address the following comments.

Major comments

There are many publications addressing the topic of allergen immunotherapy in asthma, therefore the authors need to find the gaps within the existing review papers and highlight the novelty of their proposed work (e.g. https://doi.org/10.1111/pai.13161, https://doi.org/10.3390/children7060058, https://doi.org/10.1016/j.smim.2019.101334). It was expected that this manuscript could provide the most recent overview on allergen immunotherapy, but in fact, the manuscript relies on quite old literature, especially when referring to experimental data, as evidenced in lines 70-71, 96-98, 176-179, 199-201. Additionally, it does not provide a critical overview on the clinical outcomes of allergen immunotherapy and it does not correlate the clinical outcomes with the potential biomarkers. Addressing these major points would increase the impact of the manuscript.

Minor comments:

Lines 114-115 - the sentence is difficult to follow. Do you mean "... to confirm that sensitisation is caused by HDM allergens???" Because as it is, the sentence makes no sense.

Lines 125-127 - again this sentence is not well constructed. Do you mean "The effect will be low in patients who are sensitized to haired animals and live with sensitization agents (pets) or are sensitized to other perennial allergens such as fungi." Please change accordingly or similar to clarify its meaning.

Lines 152-155 - Too redundant. Please rephrase.

Table 1 does not bring any additional information to the manuscript. Substitute this by a figure elucidating the effect of immunotherapy in the molecular markers, or something similar.

Title should be changed. It is too broad and there are already reviews with the same title.

Author Response

Response to Reviewers’ comments

Reviewer #2

The manuscript “Allergen immunotherapy in asthma” addresses the effect of allergen immunotherapy and its role in treating bronchial asthma compared to with symptomatic treatments. The manuscript is in general well written, although some sentences need to be clarified/rephrased. The proposed manuscript would be of interest to Allergies journal after the authors address the following comments.

Major comments

There are many publications addressing the topic of allergen immunotherapy in asthma, therefore the authors need to find the gaps within the existing review papers and highlight the novelty of their proposed work (e.g. https://doi.org/10.1111/pai.13161, https://doi.org/10.3390/children7060058, https://doi.org/10.1016/j.smim.2019.101334). It was expected that this manuscript could provide the most recent overview on allergen immunotherapy, but in fact, the manuscript relies on quite old literature, especially when referring to experimental data, as evidenced in lines 70-71, 96-98, 176-179, 199-201. Additionally, it does not provide a critical overview on the clinical outcomes of allergen immunotherapy and it does not correlate the clinical outcomes with the potential biomarkers. Addressing these major points would increase the impact of the manuscript.

We thank the reviewer for these helpful suggestions. We agree with the reviewer’s comments and removed most of the literature published before 2000. We also added a critical review about the effect of allergen immunotherapy on the treatment of asthma or other allergic diseases and the effect of allergen immunotherapy on the development of new allergen sensitization in the revised manuscript. We also added a description of studies that indicated a discrepancy between clinical outcomes and potential biomarkers. Furthermore, we changed the title in the revised manuscript.

Minor comments:

Lines 114-115 - the sentence is difficult to follow. Do you mean "... to confirm that sensitisation is caused by HDM allergens???" Because as it is, the sentence makes no sense.

We agree with the reviewer’s comments. We removed this sentence in the revised manuscript.

Lines 125-127 - again this sentence is not well constructed. Do you mean "The effect will be low in patients who are sensitized to haired animals and live with sensitization agents (pets) or are sensitized to other perennial allergens such as fungi." Please change accordingly or similar to clarify its meaning.

Because there is no scientific evidence for this statement, we removed this sentence from the revised manuscript.

Lines 152-155 - Too redundant. Please rephrase.

According to the suggestion, we shortened the description to make it easier to understand.

Table 1 does not bring any additional information to the manuscript. Substitute this by a figure elucidating the effect of immunotherapy in the molecular markers, or something similar.

As suggested, we added a new figure showing the effect of allergen immunotherapy on molecular markers or potential biomarker in the revised manuscript

Title should be changed. It is too broad and there are already reviews with the same title.

The title has been changed as recommended.

Round 2

Reviewer 1 Report

No other comments from me. 

Reviewer 2 Report

Accept.